# Characterization of the TCR β Chain CDR3 Repertoire in Subarachnoid Hemorrhage Patients with Delayed Cerebral Ischemia

**DOI:** 10.3390/ijms21093149

**Published:** 2020-04-29

**Authors:** Bong Jun Kim, Dong Hyuk Youn, Youngmi Kim, Jin Pyeong Jeon

**Affiliations:** 1Institute of New Frontier Stroke Research, Hallym University College of Medicine, Chuncheon 24253, Korea; luckykbj@naver.com (B.J.K.); zk61326@naver.com (D.H.Y.); kym8389@hanmail.net (Y.K.); 2Genetic and Research Inc., Chuncheon 24253, Korea; 3Department of Neurosurgery, Hallym University College of Medicine, Chuncheon 24253, Korea

**Keywords:** subarachnoid hemorrhage, delayed cerebral ischemia, T cell receptor β

## Abstract

Little is known of the adaptive immune response to subarachnoid hemorrhage (SAH). This study was the first to investigate whether T cell receptor (TCR) immune repertoire may provide a better understanding of T cell immunology in delayed cerebral ischemia (DCI). We serially collected peripheral blood in five SAH patients with DCI. High-throughput sequencing was used to analyze the TCR β chain (TCRB) complimentary determining regions (CDR) 3 repertoire. We evaluated the compositions and variations of the repertoire between admission and the DCI period, for severe DCI and non-severe DCI patients. Clonality did not differ significantly between admission and DCI. Severe DCI patients had significantly lower clonality than non-severe DCI patients (*p* value = 0.019). A read frequency of 0.005% ≤ – < 0.05% dominated the clonal expansion in non-severe DCI patients. Regarding repertoire diversity, severe DCI had a higher diversity score on admission than non-severe DCI. The CDR3 lengths were similar between admission and DCI. Among 728 annotated V-J gene pairs, we found that the relative frequencies of two V-J pairs were different at the occurrence of DCI than at admission, with T cells increasing by over 15%. TCRB CDR3 repertoires may serve as biomarkers to identify severe DCI patients.

## 1. Introduction

T cells are categorized into two groups according to the T cell receptor (TCR) expressing either α and β chains or γ and δ chains [1], and approximately 90% to 95% of the T cells in peripheral blood encode α and β chains. Every chain is composed of variable (V) and consistent (C) regions and cytoplasmic and transmembrane domains [2]. The V region of the TCR α and β chains has three complimentary determining regions (CDRs), CDR1, CDR2 and CDR3. Among these, CDR3 is the most variable of the regions, and it associated with the antigen-binding specificity of TCRs [2]. CDR3 exhibits unique TCRs through a random somatic recombination of the Variable (V), Diversity (D) and Joining (J) gene segments [1]. Over the past few years, TCR sequencing has been performed to identify immune T-cell signatures for various diseases, such as immune disease or cancer immunotherapy [3,4], suggesting the specific TCR repertoire plays a possible role as markers monitoring the immune status or treatment response [2]. In addition to immune diseases, TCR sequencing may also monitor the cardiovascular-immune microenvironment and disease pathogenesis [1]. Li et al. [1] reported that patients with acute myocardial infarction (AMI) exhibited specific CDR3 amino acids and reconstituted TCR immune repertoires.

Subarachnoid hemorrhage (SAH) is a type of hemorrhage stroke that results from the rupture of an intracranial aneurysm. It is a serious life-threatening neurological disorder with overall mortality ranging from 20% to 50% [5,6]. In particular, delayed cerebral ischemia (DCI) is a major contributor for poor neurologic outcomes following SAH, and it requires close monitoring to assess the clinical status and understand DCI pathogenesis [7,8,9]. Chrapusta et al. [10] reported that a high CD4^+^/CD8^+^ ratio and T cell-extracellular matrix interactions were associated with DCI. Ayer et al. [11] reported that the type 2 immune response involving T-helper cells modulated by statin exhibited a neuroprotective effect in an experimental SAH model. Nevertheless, the T cell response to DCI development following SAH still needs further investigation. Here, we hypothesized that identifying the TCR immune repertoire such as clonality, diversity, and rearrangement provide a better understanding of T cell immunology in DCI pathogenesis in patients with SAH.

## 2. Results

### 2.1. Clinical Characteristics of the Study

We sequenced TCR β-chain (TCRB) CDR3 repertoires from the five SAH patients treated with coil embolization and compared repertoires between the different day of admission and DCI development, as well as between severe and non-severe DCI patients. The mean age of the patients was 56 years (range 52–60 years), and most patients were female (n = 4, 80%). The aneurysm locations were as follows: middle cerebral artery (n = 2), anterior communicating artery (n = 1), posterior communicating artery (n = 1) and basilar tip (n = 1). DCI occurs ranges from 5 to 9 days after ictus. Among them, two patients (No. 4 and 5) had severe DCI and underwent additional intra-arterial chemical angioplasty (Table 1).

### 2.2. Sequence, Clonality, and CDR3 Length of TCRβ Chain CDR3 Repertoire

We calculated the total and unique sequences, clonality, and frequency of the TCRB CDR3 repertoire in each patient. Compared to admission, three non-severe DCI patients showed increased unique sequences, while two severe DCI patients showed similar or decreased unique sequences (Table 2). Regarding clonality described in the Shannon index, there is no significant difference between admission [0.088 (0.053–0.089)] and DCI [0.046 (0.023–0.056; *p* value = 0.421]. However, severe DCI patients [0.020 (0.014–0.030)] had lower clonality than non-severe DCI patients [0.088 (0.064–0.109; *p* value = 0.019] (Figure 1). A read frequency of 0.0005% ≤ – < 0.005% dominated the clonal expansion in non-severe DCI patients. On the contrary, severe DCI patients showed similar or decreased clonal expansion of sequences of 0.0005% ≤ – < 0.005% (Appendix A). We ranked TCRB clonotypes based on their frequencies and compared the repertoire diversity using the Simpson diversity index (D) [12], indicating a score close to 1 as having a high diversity (Appendix A). Overall, the diversity indices ranged from 0.990 to 0.998 on admission and ranged from 0.991 to 0.999 in the DCI period, respectively. In particular, severe DCI patients had a higher diversity score of 0.998 and 0.999 on admission compared to non-severe DCI patients.

The length of CDR3 varied between 15 and 81 nucleotides, with a peak at 42 occurring both at admission and DCI. The five most frequently observed CDR3 lengths were 36, 39, 42, 45, and 48 nucleotides. Overall, there is no significant differences in the CDR3 length ranges for patients with different time periods or DCI severity (Figure 2).

### 2.3. Distribution and Comparison of TRBV and TRBJ

We evaluated the distribution of the unique clonotypes of the TCRBV gene and TCRBJ gene. Compared to the admission day, patients with a DCI period exhibited different distribution of clonotypes in TCRBV gene segments, but similar distribution in the TCRBJ gene segments (Figure 3). We further compared the relative frequencies of the V-J combination using the difference in the frequencies of the different periods. We obtained 728 annotated V-J pairs. Among them, two V-J pairs were different, expressing a T cell increase greater than 15% in the day of DCI compared to the day of admission (Figure 4).

## 3. Discussion

Most studies on the immune reaction of stroke patients have focused on the immediate immune response after ischemic insults. Cerebral ischemia can release proinflammatory cytokines and can loosen the endothelial junctional protein, which results in an aggravation of leukocyte infiltration [13]. This inflammatory cascade is further activated by the innate immune response in brain parenchyma in a short time window. Theoretically, adaptive immunity requires a few days to allow the identification of specific antigen and clonal expansion [14]. Since DCI usually occurs after 4 days following SAH, we focused on the immune repertoire of T cells that recognize antigens from the major histocompatibility complex using TCR [2]. Previously, the TCR immune repertoires were reported as a novel method to monitor and assess T lymphocyte in patients with AMI [1]. Therefore, we identified and tracked the composition and variation of the TCRB CDR3 repertoires in SAH patients, particularly focusing on the DCI development. In our study, there is no significant difference in the clonality, CDR3 length and rearrangement between the day of admission and DCI. However, severe DCI patients showed significantly decreased clonality and higher diversity than non-severe DCI patients.

T cells are known to be related to stroke pathogenesis, especially during the early stroke period as the key mediators of adaptive immunity [14]. In general, SAH is thought to probably stimulate innate and adaptive immune reactions [15]. However, only few studies have attempted to determine the adaptive immune response in SAH patients. Mathiesen et al. [16] measured soluble CD8^+^ levels in SAH patients. In their study, higher serum soluble CD8^+^ was noted in four SAH patients (66.7%) on day 3 after ictus. However, a definite association between soluble CD8^+^ and DCI has not been observed. Ayer et al. [11] evaluated the immune modulating effect of statin in a rat model of SAH. The rat treated with a high dose of simvastatin showed a higher expression of immune suppressive cytokine T-regulatory transforming growth factor-β1, indicating the potential role statin plays in modulating the type 2 immune response with T-helper cells. Moraes et al. [15] measured the leukocyte subpopulation in 12 SAH patients. They showed a marked activation of the innate immune cells, such as monocytes (CD14^++^ and CD16^+^) or neutrophils (CD69^+^) in the CSF. In addition, SAH patients had activated immune cells such as higher CD4^+^ and CD8^+^ T cells in the CSF. Nevertheless, the effect of early immune activation on adaptive immune system remains undetermined in SAH patients, requiring further studies.

We performed the first study to compare TCR repertoire using high-throughput sequencing in SAH patients with DCI. In previous studies, features of TCR repertoire have been widely studied within the context of autoimmune disease, transplantation, or cancers with immunotherapy. Chang et al. [12] evaluated TCRB repertoire of circulating T lymphocytes in eight rheumatoid arthritis patients who were treated with different biologic medications. The results showed an inverse tendency between the disease activity and TCRB repertoire diversity. Sakurai et al. [17] also showed that the repertoire diversity of memory CD4^+^ T cells was decreased in rheumatoid arthritis patients and that shared epitope alleles were reduced compared to healthy controls. In addition, TCR repertoire diversity was negatively associated with both shared epitope allele dosage and disease activity. Therefore, they concluded that restoring the altered TCR repertoire diversity represent a potential therapeutic target for rheumatoid arthritis. Yang et al. [2] examined TCRB CDR3 repertoire in six liver transplantation patients with different time periods, such as pre-transplantation and the first and seventh days after transplantation. Although the lengths of CDR3, VD indel and DJ indel were similar among groups, more expanded T cell clones were observed in patients in the pre-transplantation period than those in other time periods. Regarding cancer patients who underwent immunotherapy, similar findings were observed. Hopkins et al. [18] sequenced the TCR repertoire for metastatic pancreatic cancer patients treated with different biologic medications. The results showed that lower baseline peripheral TCR clonality and higher expansion were associated with longer survival. In addition, different biologic medications showed different effects on the peripheral repertoire. Accordingly, they suggested that profiling the TCR repertoire can serve as a biomarker for clinical response in cancer patients who received immunotherapy. Beyond immunocompromised patients, the diversity of TCR also can be changed following AMI [1]. Specifically, AMI patients have shown diminished diversity of CDR3 amino acid compared to healthy controls. By contrast, in our study, severe DCI patients showed a marked decrease in TCRB clonality as well as a higher diversity compared to non-severe DCI patients. In addition, the proportion of shared CDR3 amino acids was lower in severe DCI patients than that in non-severe DCI patients (Appendix A). We believe that these conflicting results can be attributed to differences in disease characteristics. Most previous studies have compared TCRB CDR3 repertoires between disease patients and healthy controls [1,12,17,18]. However, we evaluated TCRB CDR3 repertoires in SAH patients presenting with decreased consciousness as well as neurologic deficits. In addition, we have compared TCRB CDR3 repertoires based on DCI severities. In clinical settings, early detection of severe DCI refractory to medical treatments and subsequent endovascular intervention is of main concern. Nevertheless, differences in DCI pathogenesis, particularly between severe and non-severe DCI, have yet to be investigated in detail. Based on our results, monitoring TCR immune repertoires could be a marker for identifying severe DCI patients who are at risk of poor neurologic outcomes in the neurointensive care units.

There are some limitations in our investigation. First, the small sample size may limit the statistical power. In most previous studies, the number of enrolled patients was fewer than ten due to the high experimental costs of sequencing TCRB [12,19]. Nevertheless, we observed a correlation between DCI severity and the diversity and clonality of TCRB repertoires; thus, we expect that an upcoming study will validate our findings with a large number of SAH patients. Second, we enrolled SAH patients who underwent endovascular coil embolization. Accordingly, SAH patients who had surgical clipping may have different TCRB repertoires in the DCI pathogenesis. Third, age can bias the TCRB repertoires. An older age was related to a decrease in the repertoire diversity in patients with rheumatoid arthritis, irrespective of different medications [12]. Although, the mean age of the severe DCI and non-severe DCI patients in our study were 54.5 and 57, respectively, additional studies of TCRB repertoire considering age and treatment method are required.

In conclusion, our results suggest the possible role of an adaptive immune response in severe DCI pathogenesis. Monitoring the TCRB CDR3 repertoires could be useful to identify severe DCI patients who are refractory to medical treatments. More research is required to understand the adaptive immunity of severe DCI and its therapeutic targets.

## 4. Materials and Methods

### 4.1. Study Population

The study cohort was obtained from the Chuncheon Sacred Heart Hospital stroke database between March 2016 and June 2019. This database results from an ongoing prospective, observational project in the regional medical center of the district of Chuncheon City, the capital city of Gangwon Province in Korea [20,21,22,23]. In this database, we selected five spontaneous SAH patients with following conditions: (1) are adults over 18 years old; (2) presented a saccular aneurysm; (3) underwent coil embolization; and (4) had blood samplings taken at admission and at the DCI period.

The primary outcome was to evaluate the composition and variations of the TCRB CDR3 repertoire between the two different time periods of admission and DCI. The secondary outcome was to compare the TCRB CDR3 repertoire according to the DCI severity (severe DCI vs. non-severe DCI). DCI was diagnosed when patients showed new neurologic symptoms including dysphasia, motor weakness, sensory change, and decreased consciousness with concomitant severe cerebral vasospasm [22,24]. We monitored DCI daily using transcranial Doppler ultrasonography. When DCI was suspected, a CT-angiography was performed to assess the degree of vasospasm, followed by the maintenance of hypertension and hypervolemia. DCI was divided into two groups of severe and non-severe DCI. Severe DCI was defined as that where patients underwent chemical angioplasty to reverse vasospasm refractory to medical treatment. SAH patients who underwent Medical information (e.g., age, hypertension, diabetes mellitus, hyperlipidemia, and smoking) and radiologic findings (e.g., Hunt and Hess grade, Fisher grade, aneurysm location and size) was reviewed. This study was approved by the Institutional Review Board at the participating hospital (No. 2016-3, 2017-9, 2018-6 and 2019-6), and informed consent was received from the patients or their relatives.

### 4.2. DNA Extraction

Peripheral blood samples were obtained at the day of admission (within three hours after admission) and upon DCI development (within three hours after the occurrence of DCI symptoms). Genomic DNA was prepared from whole blood using the QIAamp DNA Blood Midi Kit (Qiagen, Hilden, Germany) according to the manufacturer’s instructions. In brief, whole blood samples were collected in EDTA tubes. After centrifugation, the buffy coat layer was harvested and genomic DNA was extracted. The samples were quantified using Dropsense96 and diluted for library preparation in buffer to standard.

### 4.3. High-Throughput Sequencing of the CDR3 Region

The sample data was generated using the immunoSEQ assay (Adaptive Biotechnologies, Seattle, WA). The somatically rearranged locus CDR3 region was amplified from genomic DNA using bias-controlled multiplex polymerase chain reaction (PCR) amplification [25,26]. Specifically, the first PCR consists of forward and reverse amplification primers specific for every V and J gene segment, and it amplifies the hypervariable CDR3 of the immune receptor locus. The second PCR adds a proprietary barcode sequence (Adaptive Biotechnologies, Seattle, WA, USA) and Illumina adapter sequences (Illumina, San Diego, CA, USA) [26,27]. CDR3 libraries were sequenced on an Illumina instrument according to the manufacturer’s instructions.

### 4.4. Analysis of the TCRβ Chain CDR3 Repertoire

The raw Illumina sequence reads were demultiplexed according to Adaptive’s proprietary barcode sequences. The demultiplexed reads were then further processed to remove adapter and primer sequences; identify and correct for technical errors introduced through PCR and sequencing; and remove primer dimer, germline and other contaminant sequences. The data is filtered and clustered using both the relative frequency ratio between similar clones and a modified nearest-neighbor algorithm to merge closely-related sequences. The resulting sequences were sufficient to allow annotation of the V(N)D(N)J genes constituting each unique CDR3 and the translation of the encoded CDR3 amino acid sequence. V, D and J gene definitions were based on annotations in accordance with the IMGT database. The set of observed biological locus CDR3 sequences were normalized to correct for residual multiplex PCR amplification bias and quantified against a set of synthetic locus CDR3 sequence analogues [26]. Data was analyzed using the immunoSEQ Analyzer toolset.

## Figures and Tables

**Figure 1 ijms-21-03149-f001:**
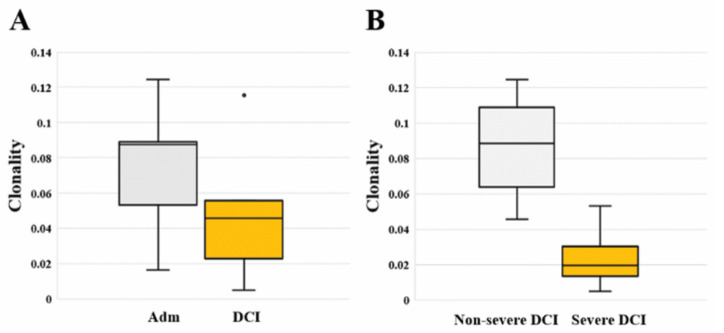
(**A**) Comparison of TCRB clonality between the two different time periods of the day of admission (Adm) and delayed cerebral ischemia (DCI). Clonality did not differ significantly between admission [0.088 (0.053–0.089)] and DCI [0.046 (0.023–0.056); *p* value = 0.421]. (**B**) Comparison of TCRB clonality between severe DCI and non-severe DCI. Clonality of severe DCI patients [0.020 (0.014–0.030)] was significantly higher than that of non-severe DCI patients [0.088 (0.064–0.109; *p* value = 0.019].

**Figure 2 ijms-21-03149-f002:**
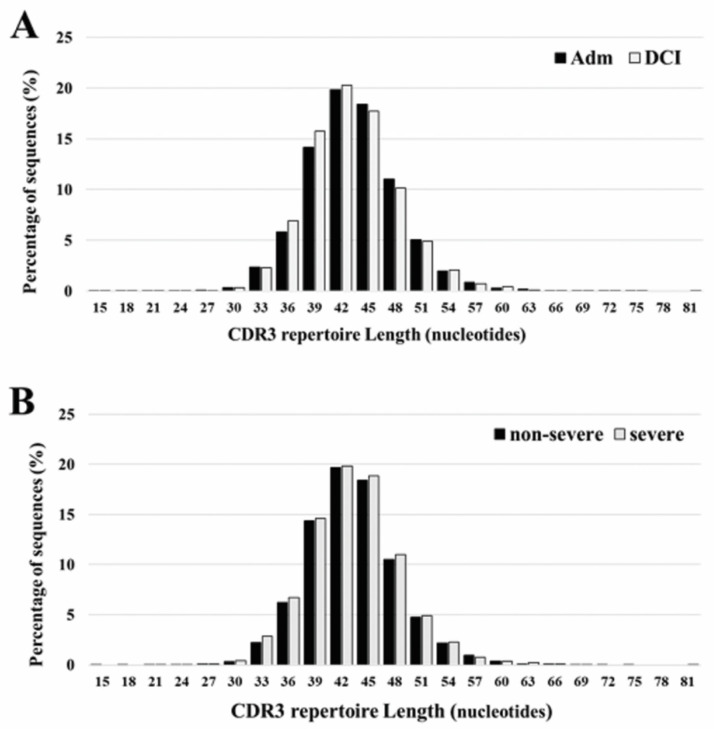
CDR3 repertoire length distribution of TCRB between admission (Adm) and delayed cerebral ischemia (DCI) (**A**), and severe DCI and non-severe DCI (**B**). The five most frequently observed CDR3 lengths were 36, 39, 42, 45, and 48 nucleotides. The results are the average of five individuals in each group.

**Figure 3 ijms-21-03149-f003:**
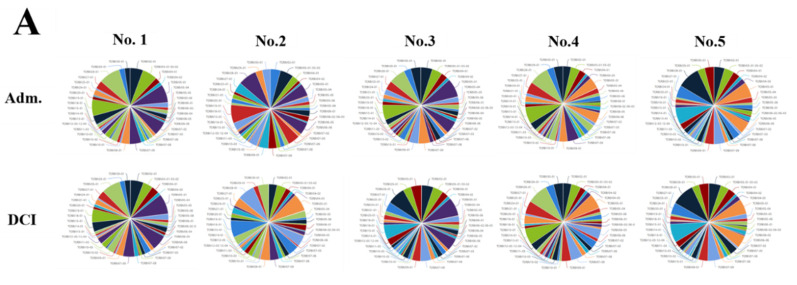
Distribution of the unique clonotypes of TCRBV gene (**A**) and TCRBJ gene (**B**) in patients with subarachnoid hemorrhage between the day of admission and delayed cerebral ischemia (DCI). Different colors represent differences in the gene frequencies of the TCRBV and TCRBJ genes. Compared to the TCRBJ gene segments, the TCRBV gene segments in DCI patients showed different clonotypes in the different time periods.

**Figure 4 ijms-21-03149-f004:**
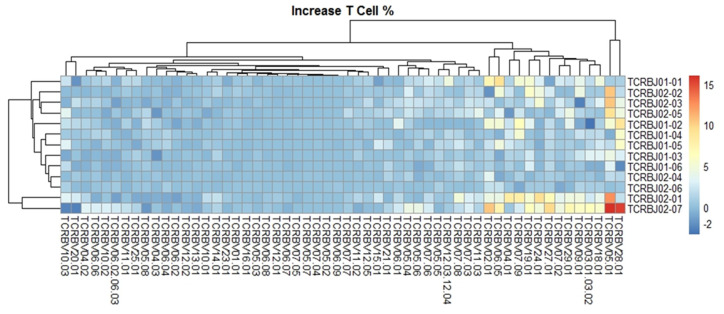
Comparison of the average TCRB V-J gene utilization between the day of admission and delayed cerebral ischemia (DCI). V gene and J gene segments are arranged on the x-axis and the y-axis, respectively. Different colors represent differences in the mean gene frequencies between the two different periods. Among 728 annotated V-J pairs, two V-J pairs, TCRBV28-01/TCRBJ02-07 and TCRBV05-01/TCRBJ02-07, were increased by more than 15% from the day of admission to DCI.

**Table 1 ijms-21-03149-t001:** Baseline characteristics of the enrolled patients. All patients underwent coil embolization within 6 h after the onset of subarachnoid hemorrhage.

Pt No.	Age	Sex	Hunt and Hess Grade	Fisher Grade	Days of DCI	Aneurysm Site	Chemical Angioplasty	Underlying Medical Conditions
1	58	F	III	3	8	BA	No	Hypertension
2	60	M	IV	3	9	A-com	No	Diabetes mellitus
3	53	F	III	4	5	P-com	No	Hypertension, hyperlipidemia
4	52	F	IV	4	8	MCA	Yes *	Hypertension, hyperlipidemia
5	57	F	IV	4	7	MCA	Yes *	Hypertension

A-com, anterior communicating artery; BA, basilar artery; DCI, delayed cerebral ischemia; MCA, middle cerebral artery; P-com, posterior communicating artery; Pt No., patient number; * indicates the patients with severe DCI refractory to medical treatments.

**Table 2 ijms-21-03149-t002:** TCRB CDR3 repertoire with total sequences, unique sequences, and clonality.

Pt		gDNA ^1^	N.A ^2^			Frame	Sequences (A.A.) ^3^	
No.	Group	Amount (ng)	Total	Unique	Total	Unique	Clonality	Max Frequency (%)
1	Adm	670.38	4590	3104	3747	2464	0.087	4.29
	DCI	670.40	5125	3497	4186	2785	0.045	2.01
2	Adm	666.78	1614	1079	1270	860	0.089	5.27
	DCI	465.30	1879	1402	1527	1148	0.056	3.41
3	Adm	668.64	7327	4623	5829	3652	0.124	7.92
	DCI	665.62	15670	10163	12555	8073	0.116	6.89
4	Adm	668.07	5773	4918	4804	4120	0.017	0.97
	DCI	665.12	5387	4955	4498	4128	0.005	0.28
5	Adm	717.79	7428	5650	6177	4576	0.053	1.95
	DCI	756.19	5072	4324	4238	3569	0.023	0.97

^1^ GenomicDNA; ^2^ The number of nucleic acid (N.A.) sequences; ^3^ The number of amino acid (A.A.) sequences.

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
