# Peer review of "Characterization of the TCR β Chain CDR3 Repertoire in Subarachnoid Hemorrhage Patients with Delayed Cerebral Ischemia"

_ijms, 2020, doi:10.3390/ijms21093149_

Round 1
Reviewer 1 Report
This is interesting and important study on TCR CD3 clonality as a novel biomarker. There are minor points to address:
- For Materials and Methods-describe in more details with scheme of nested PCR as a supplemental Figure-in order to understand how sequencing of CD3 was performed.
- In Results this sentences the first sentence needs to be corrected:"The length of CDR3 varied between 15 and 81 amino acids, with a peak at 42 in both at admission 91 and DCI. The five most frequently observed CDR3 lengths were 36, 39, 42, 45, and 48 nt." The length should be nucleotides not amino-acids.
- Figure 2; X-axis, include length in nucleotides
- Figure 3, explain different colors meaning in circles
- Figure 1 and 4, include better Figure description with more details
- Discussion: Please include better mechanism of clonality changes and biological meaning of the Results, importance of these findings in clinic with references of other markers.
Author Response
Comment 1: For Materials and Methods-describe in more details with scheme of nested PCR as a supplemental Figure-in order to understand how sequencing of CD3 was performed.
Answer: Thank you for your comments on our manuscript. The analytical procedure was conducted by Adaptive Biotechnologies (Seattle, WA, USA) after sampling (Figure 1, below). We asked the company to provide a more detailed procedure of the immunoSEQ assay using a supplemental figure, but the company recommended we present the reference lists shown below (References 1 and 2, below). As you know, the immunoSEQ assay combines multiple PCR with high-throughput sequencing and pipeline for CDR3 region analysis (References 1 and 3, below) (Figure 2, below). Instead of providing nested PCR in the supplemental figure, we have added in the method section a link to the URL (https://www.adaptivebiotech.com//immunoseq) that describe the immunoSEQ assay in detail.
Figure 1. A schematic summary of the study.
Figure 2. ImmunoSEQ assay of two-step PCR provided by Adaptive Biotechnologies.
References
- Carlson CS, Emerson RO, Sherwood AM, Desmarais C, Chung MW, Parsons JM, et al. Using synthetic templates to design an unbiased multiplex PCR assay. Nat Commun. 2013; 4:2680.
- Robins H, Desmarais C, Matthis J, Livingston R, Andriesen J, Reijonen H, et al. Ultra-sensitive detection of rare T cell clones. J Immunol Methods. 2012; 375:14-19.
- Robins HS1, Campregher PV, Srivastava SK, Wacher A, Turtle CJ, Kahsai O et al. Comprehensive assessment of T-cell receptor beta-chain diversity in alphabeta T cells. Blood. 2009; 114:4099-4107.
Comment 2: In Results this sentences the first sentence needs to be corrected: "The length of CDR3 varied between 15 and 81 amino acids, with a peak at 42 in both at admission 91 and DCI. The five most frequently observed CDR3 lengths were 36, 39, 42, 45, and 48 nt." The length should be nucleotides not amino-acids.
Answer: Thank you for your comments on our manuscript. We have revised the results as follows: The length of CDR3 varied between 15 and 81 nucleotides, with a peak at 42 occurring both at admission and DCI. The five most frequently observed CDR3 lengths were 36, 39, 42, 45, and 48 nucleotides.
Comment 3: Figure 2; X-axis, include length in nucleotides
Answer: Per your recommendation, we have revised the figures as follows:
Comment 4: Figure 3, explain different colors meaning in circles
Answer: We have revised the Figure 3 caption as follows:
Figure 3. Distribution of the unique clonotypes of TCRBV gene (A) and TCRBJ gene (B) in patients with subarachnoid hemorrhage between the day of admission and delayed cerebral ischemia (DCI). Different colors represent differences in the gene frequencies of the TCRBV and TCRBJ genes. Compared to the TCRBJ gene segments, the TCRBV gene segments in DCI patients showed different clonotypes in the different time periods.
Comment 5: Figure 1 and 4, include better Figure description with more details
Answer: Per your recommendation, we have revised the figures and added additional details in the descriptions as follows:
Figure 1. A, Comparison of TCRB clonality between the two different time periods of the day of admission (Adm) and delayed cerebral ischemia (DCI). Clonality did not differ significantly between admission [0.088 (0.053–0.089)] and DCI [0.046 (0.023–0.056); p=0.421]. B, Comparison of TCRB clonality between severe DCI and non-severe DCI. Clonality of severe DCI patients [0.020 (0.014–0.030)] was significantly higher than that of non-severe DCI patients [0.088 (0.064–0.109; p=0.019].
Figure 4. Comparison of the average TCRB V-J gene utilization between the day of admission and delayed cerebral ischemia (DCI). V gene and J gene segments are arranged on the x-axis and the y-axis, respectively. Different colors represent differences in the mean gene frequencies between the two different periods. Among 728 annotated V-J pairs, two V-J pairs, TCRBV28-01/TCRBJ02-07 and TCRBV05-01/TCRBJ02-07, were increased by more than 15% from the day of admission to DCI.
Comment 6: Discussion: Please include better mechanism of clonality changes and biological meaning of the Results, importance of these findings in clinic with references of other markers.
Answer: Per your recommendation, we have revised the discussion as follows:
We performed the first study to compare TCR repertoire using high-throughput sequencing in SAH patients with DCI. In previous studies, features of TCR repertoire have been widely studied within the context of autoimmune disease, transplantation, or cancers with immunotherapy. Chang et al. (Reference 1, below) evaluated the TCRB repertoire of circulating T lymphocytes in eight rheumatoid arthritis patients who were treated with different biologic medications. The results showed an inverse tendency between the disease activity and TCRB repertoire diversity. Sakurai et al. (Reference 2, below) also showed that the repertoire diversity of memory CD4+ T cells was decreased in rheumatoid arthritis patients and that shared epitope alleles were reduced compared to healthy controls. In addition, TCR repertoire diversity was negatively associated with both shared epitope allele dosage and disease activity. Therefore, they concluded that restoring the altered TCR repertoire diversity represent a potential therapeutic target for rheumatoid arthritis. Regarding cancer patients who underwent immunotherapy, similar findings were observed. Hopkins et al. (Reference 3, below) sequenced the TCR repertoire for metastatic pancreatic cancer patients treated with different biologic medications. The results showed that lower baseline peripheral TCR clonality and higher expansion were associated with longer survival. In addition, different biologic medications showed different effects on the peripheral repertoire. Accordingly, they suggested that profiling the TCR repertoire can serve as a biomarker for clinical response in cancer patients who received immunotherapy. Beyond immunocompromised patients, the diversity of TCR also can be changed following AMI (Reference 4, below). Specifically, AMI patients have shown diminished diversity of CDR3 amino acid compared to healthy controls. By contrast, in our study, severe DCI patients showed a marked decrease in TCRB clonality as well as a higher diversity compared to non-severe DCI patients. In addition, the proportion of shared CDR3 amino acids was lower in severe DCI patients than that in non-severe DCI patients. We believe that these conflicting results can be attributed to differences in disease characteristics. Most previous studies have compared TCRB CDR3 repertoires between disease patients and healthy controls (References 1-4, below). However, we evaluated TCRB CDR3 repertoires in SAH patients presenting with decreased consciousness as well as neurologic deficits. In addition, we have compared TCRB CDR3 repertoires based on DCI severities. In clinical settings, early detection of severe DCI refractory to medical treatments and subsequent endovascular intervention is of main concern. Nevertheless, differences in DCI pathogenesis, particularly between severe and non-severe DCI, have yet to be investigated in detail. Based on our results, monitoring TCR immune repertoires could be a marker for identifying severe DCI patients who are at risk of poor neurologic outcomes in neurointensive care units.
References
- Chang CM, Hsu YW, Wong HS, Wei JC, Liu X, Liao HT, et al. Characterization of T-Cell Receptor Repertoire in Patients with Rheumatoid Arthritis Receiving Biologic Therapies. Dis Markers. 2019; 2019:2364943.
- Sakurai K, Ishigaki K, Shoda H, Nagafuchi Y, Tsuchida Y, Sumitomo S, et al. HLA-DRB1 Shared Epitope Alleles and Disease Activity Are Correlated with Reduced T Cell Receptor Repertoire Diversity in CD4+ T Cells in Rheumatoid Arthritis. J Rheumatol. 2018; 45:905-914.
- Hopkins AC, Yarchoan M, Durham JN, Yusko EC, Rytlewski JA, Robins HS, et al. T cell receptor repertoire features associated with survival in immunotherapy-treated pancreatic ductal adenocarcinoma. JCI Insight. 2018; 3(13).
- Li D, Hu L, Liang Q, Zhang C, Shi Y, Wang B, et al. Peripheral T cell receptor beta immune repertoire is promptly reconstituted after acute myocardial infarction. J Transl Med. 2019; 17:40.

Reviewer 2 Report
The manuscript entitled "Characterization of TCR β-chain CDR3 Repertoire in
Subarachnoid Hemorrhage Patients with Delayed Cerebral Ischemia" by Kim et al inherited about the evaluation of TRC β-chain rearrangements in Subarachnoid Hemorrhage Patients with Delayed Cerebral Ischemia is well written and suitable for pubblication after minor revisions:
- Please, could the authors better express criteria to perform blood withdrawn? which are the aspects that regulate this step?
- In the section material and methods, please could the authors define which was the cfDNA input to generate libraries?
- Please, could the authors better describe the NGS platform that they adopted ?
- Please, could the authors better discuss the low number of patients of this study? In my opinion more patients should be required to validate this idea
Author Response
Comment 1: Please, could the authors better express criteria to perform blood withdrawn? which are the aspects that regulate this step?
Answer: Thank you for your comments on our manuscript. We have revised the DNA extraction section as follows:
Peripheral blood samples were obtained at the day of admission (within three hours after admission) and upon DCI development (within three hours after the occurrence of DCI symptoms). Genomic DNA was prepared from whole blood using the QIAamp DNA Blood Midi Kit (Qiagen, Hilden, Germany) according to the manufacturer’s instructions. In brief, whole blood samples were collected in EDTA tubes. After centrifugation, the buffy coat layer was harvested and genomic DNA was extracted. The samples were quantified using Dropsense96 and diluted for library preparation in buffer to standard.
Comment 2: In the section material and methods, please could the authors define which was the cfDNA input to generate libraries?
Answer: Thank you for your comment on our manucript. A schematic summary of the study is shown in the Figure below. After genomic DNA (gDNA) extraction, immunoSEQ assay was performed. For the sake of clarity, we have included the amount of gDNA in Table 2, below.
Figure 1. A schematic summary of the study.
Table 2. TCRB CDR3 repertoire with total sequences, unique sequences, and clonality.
|
Pt |
|
gDNA1 amount (ng) |
N.A2 |
Frame Sequences (A.A.)3 |
|||||
|
No. |
Group |
Total |
Unique |
Total |
Unique |
Clonality |
Max frequency (%) |
||
|
1 |
Adm |
670.38 |
4590 |
3104 |
3747 |
2464 |
0.087 |
4.29 |
|
|
|
DCI |
670.40 |
5125 |
3497 |
4186 |
2785 |
0.045 |
2.01 |
|
|
2 |
Adm |
666.78 |
1614 |
1079 |
1270 |
860 |
0.089 |
5.27 |
|
|
|
DCI |
465.30 |
1879 |
1402 |
1527 |
1148 |
0.056 |
3.41 |
|
|
3 |
Adm |
668.64 |
7327 |
4623 |
5829 |
3652 |
0.124 |
7.92 |
|
|
|
DCI |
665.62 |
15670 |
10163 |
12555 |
8073 |
0.116 |
6.89 |
|
|
4 |
Adm |
668.07 |
5773 |
4918 |
4804 |
4120 |
0.017 |
0.97 |
|
|
DCI |
665.12 |
5387 |
4955 |
4498 |
4128 |
0.005 |
0.28 |
||
|
5 |
Adm |
717.79 |
7428 |
5650 |
6177 |
4576 |
0.053 |
1.95 |
|
|
|
DCI |
756.19 |
5072 |
4324 |
4238 |
3569 |
0.023 |
0.97 |
|
1GenomicDNA;2 The number of nucleic acid (N.A.) sequences; 3The number of amino acid (A.A.) sequences.
Comment 3: Please, could the authors better describe the NGS platform that they adopted?
Answer: Multiplex PCR primers contained the gene specific sequences on the 3′-end and the universal adaptor sequences on the 5′-end. Low-cycle PCR was used to integrate Illumina sequencing adaptors, and the resultant amplification was sequenced on the Illumina MiSeq instrument using V gene forward and J gene reverse sequencing primers, while the second internal barcode was used to precisely identify each template (References 1 and 2, below).
References
- Carlson CS, Emerson RO, Sherwood AM, Desmarais C, Chung MW, Parsons JM, et al. Using synthetic templates to design an unbiased multiplex PCR assay. Nat Commun. 2013; 4:2680.
- Robins H, Desmarais C, Matthis J, Livingston R, Andriesen J, Reijonen H, et al. Ultra-sensitive detection of rare T cell clones. J Immunol Methods. 2012; 375:14-19.
Comment 4: Please, could the authors better discuss the low number of patients of this study? In my opinion more patients should be required to validate this idea
Answer: Per your recommendation, we have included a discussion of the low number of patients in the study as a limitation, which reads as follows:
There are some limitations in our investigation. First, the small sample size may limit the statistical power. In most previous studies, the number of enrolled patients was fewer than ten due to the high experimental costs of sequencing TCRB (References 1 and 2, below). Nevertheless, we observed a correlation between DCI severity and the diversity and clonality of TCRB repertoires; thus, we expect that an upcoming study will validate our findings with a large number of SAH patients.
References
- Chang CM, Hsu YW, Wong HS, Wei JC, Liu X, Liao HT, et al. Characterization of t-cell receptor repertoire in patients with rheumatoid arthritis receiving biologic therapies. Dis Markers. 2019;2019:2364943
- Hou X, Wang M, Lu C, Xie Q, Cui G, Chen J, et al. Analysis of the repertoire features of tcr beta chain cdr3 in human by high-throughput sequencing. Cell Physiol Biochem. 2016;39:651-667

Round 2
Reviewer 2 Report
The manuscript is suitable for pubblication without any revisions